# Impact of time and temperature on gut microbiota and SCFA composition in stool samples

**Janet L. Cunningham**[1], **Ludvig Bramstång**[1], **Abhijeet Singh**[2], **Shishanthi Jayarathna**[2], **Annica J. Rasmusson**[1], **Ali Moazzami**[2], **Bettina Müller**[2]*

**1** Department of Neurosciences, Psychiatry, Uppsala University, Uppsala, Sweden, **2** Department of Molecular Sciences, BioCentrum, Swedish University of Agricultural Sciences, Uppsala, Sweden

* Bettina.Muller@slu.se

**Data Availability Statement:** The sequencing raw data has been submitted to NCBI sequence read archive (SRA) under the BioProject accession number PRJNA609715.

## Abstract

Gut dysbiosis has been implicated in the pathophysiology of a growing number of non-communicable diseases. High through-put sequencing technologies and short chain fatty acid (SCFA) profiling enables surveying of the composition and function of the gut microbiota and provide key insights into host-microbiome interactions. However, a methodological problem with analyzing stool samples is that samples are treated and stored differently prior to submission for analysis potentially influencing the composition of the microbiota and its metabolites. In the present study, we simulated the sample acquisition of a large-scale study, in which stool samples were stored for up to two days in the fridge or at room temperature before being handed over to the hospital. To assess the influence of time and temperature on the microbial community and on SCFA composition in a controlled experimental setting, the stool samples of 10 individuals were exposed to room and fridge temperatures for 24 and 48 hours, respectively, and analyzed using 16S rRNA gene amplicon sequencing, qPCR and nuclear magnetic resonance spectroscopy. To best of our knowledge, this is the first study to investigate the influence of storage time and temperature on the absolute abundance of methanogens, and of *Lactobacillus reuteri*. The results indicate that values obtained for methanogens, *L. reuteri* and total bacteria are still representative even after storage for up to 48 hours at RT (20˚C) or 4˚C. The overall microbial composition and structure appeared to be influenced more by laboratory errors introduced during sample processing than by the actual effects of temperature and time. Although microbial activity was demonstrated by elevated SCFA at both 4˚C and RT, SCFAs ratios were more stable over the different conditions and may be considered as long as samples are come from similar storage conditions.

## Introduction

The gut microbiota (GM) has become of increasing interest as gut dysbiosis has been implicated in the pathophysiology or exacerbation of a growing number of non-communicable

**Funding:** Funding for this study was obtained by Janet Cunningham from the Ekhaga foundation (http://www.ekhagastiftelsen.se), The Swedish Society of Medicine (https://www.sls.se); and ALF Funds from Uppsala University Hospital (https://www.akademiska.se). The funders had no role in study design, data collection and analysis, decision to publish, or preparation of the manuscript.

**Competing interests:** The authors have declared that no competing interests exist.

diseases including diabetes mellitus, obesity, allergies, rheumatoid arthritis, inflammatory bowel disease, liver disease, colorectal cancer, Parkinson's disease, Alzheimer's disease, multiple sclerosis, autism spectrum disorder, depression and anxiety disorders [1–7]. The gut-brain-axis pertains to the bidirectional communication between gut microbiota and the central nervous system through neural, endocrine, immunological and metabolic means. Short-chain fatty acids (SCFAs) are the main metabolic products from bacterial fermentation in the intestine, and have a key role in microbiota-gut-brain cross talk [8]. The human microbiota consists of at least 1 000 species of bacteria with varying compositions and densities at different sites, as well as protozoa, viruses and fungi [9]. DNA sequencing technologies including 16S rRNA gene based amplicon sequencing is the main approach used to study microbial diversity, and to understand the role of the gut microbiome in human health and disease. A methodological problem with analyzing stool samples, however, is that the logistics of collecting samples can vary dramatically between the subjects which may influence the composition of the microbiota and its metabolic products. The current understanding is that time and temperature appears to have a low, but not negligible impact on bacterial composition and structure in stool samples [10–15]. Immediate freezing of samples at -20°C is considered to be the "gold standard", but might not always be practically feasible. A number of fecal collection methods including immediate addition of preservation solutions such as ethanol or RNA stabilizers or direct application of commercial kits such as the OMNIgene•GUT have been tested and found to be comparable to the reference of instant freezing at least when analyzing more abundant taxa [11–13, 15, 16]. Although most of these collection methods are "participant-friendly", participants are not always able to follow the instructions. Moreover, a vast amount of stool samples may have been collected and stored in repositories without any immediate preservation. In an ongoing large-scale study, we are aiming to explore the role of the gut microbiota and SCFAs in depression and mood disorders. We have, however, observed differences between the patients in terms of storage conditions before the time point for sample submission. To estimate to which degree different management of samples prior to freezing and without preservation might influence the results of future studies, the effect of different temperatures over time on microbial composition and SCFA profile of stool samples needs to be further investigated.

The evidence in the literature on this topic is very limited and, to our knowledge, there is no study done regarding this matter relating specifically to low abundant groups such as methanogens and Lactobacilli. *Lactobacillus* spp. are found in the gastrointestinal systems in variable amounts depending on species and age and has gained increasing attention due to their probiotic attributes [17–19]. The role of methanogens in human health and the possible association to infections and diseases are still poorly understood but a mounting body of evidences suggest a significant role [20–24]. Methanogens play an important role in removing excess hydrogen gas from the gut and improving efficiency of microbial fermentation, but also alter the SCFA production [25]. The impact of time and temperature on the SCFA concentration in stool samples, collected without an immediate preservation method, has been insufficiently studied. Recent studies reported rapid changes of the SCFA level in stool samples stored at RT and several human fecal sample collection protocols have been proposed [15, 26, 27]. So far, no conclusions have been drawn for the consideration of SCFA data originating from stool specimens that could not be obtained in accordance with proposed protocols.

The primary aim of this methodological study was to analyze the impact of time and temperature on the absolute abundance of methanogens, *Lactobacillus reuteri*, as well as total bacteria, overall microbial composition and the SCFA profile. This would facilitate the determination of reliable thresholds that can be potentially useful in larger studies, where uniform sample collection conducted at home cannot be guaranteed. For doing so, we have

chosen times and temperatures for sample storage before being handed-over to the hospital that are within the range of those reported by some participants in our large-scale studies.

## Material and methods

### Ethics and subject recruitment

The study includes ten subjects. To facilitate the collection of fresh samples, we recruited five patients from a psychiatric ward where they were undergoing treatment for affective disorders. Samples from patients were collected from the Uppsala Psychiatric Patient Samples (UPP) Framework approved by the Regional Ethics Committee in Uppsala: (Dnr 2012/81, 2012-03-21, Dnr 2012/81/1, 2012-12-20, Dnr 2013/219). Written informed consent was obtained from patients for material collection. The results from these samples are not linked to individual factors in this study. Additionally, five medical students without any specific inclusion or exclusion criteria participated voluntary and without any compensation. They contributed to this study anonymously and their data cannot be tied to specific subjects. They gave verbal informed consent for their donated samples to be analyzed anonymously with the purpose of method development and neither samples nor data can be traced back to control individuals. The Regional Ethics Committee waived the need for consent in this case in accordance with Swedish law.

### Sample management

The subjects were given instructions to store their sample in fridge or cooler immediately upon acquiring and then initiate contact for transportation. Samples were collected within 4 hours and transported at $< 4\degree$C using a cooling bag with ice clamps. After arrival at the laboratory, the samples were mechanically homogenized with a sterile spatula, aliquoted into five Nunc Cryotube Vials™ and entered into the storing protocol including five storage conditions: the first aliquot was frozen immediately at -20°C (1), the second was frozen after keeping it 24 hours at room temperature (RT) (20–21°C) (2), the third after 48 hours at room temperature (3), the fourth after 24 hours at 4°C (4), and the fifth was frozen after 48 hours at 4°C (5). DNA was extracted from all frozen samples within 48–96 hours after completion of the storing protocol.

### DNA purification

DNA was extracted using the QIAamp$^{©}$ Fast DNA Stool Mini Kit including an additional bead-beating step: 200±1 mg stool was weighed in Lysing Matrix E tubes (MPBiomedicals™) and placed on ice. One ml of InhibitEX Buffer was added and the sample was vortexed continuously for 1 min. The tube was placed in a FastPrep Instrument (MPBiomedicals™) for 40 seconds at speed setting 6.0, and centrifuged at 14 000 x g for 10 minutes. Then, 600 μl of the obtained supernatant was transferred into a new microcentrifuge tube and treated as described in the manufacturer´s protocol. The DNA concentration was approximated using Qubit$^{©}$ fluorometer dsDNA protocol (Invitrogen). DNA was purified in triplicates from every storage condition except for three samples due to insufficient material. Out of these three, one was purified in duplicates for all conditions (sample B), and two were purified in triplicates but covering only storage condition 1, 4 and 5 (sample I, and J).

### Quantitative PCR (qPCR)

Absolute quantification using qPCR was performed for total bacteria, methanogenic archaea, and *Lactobacillus reuteri*. Methanogens were quantified using the group specific primers

Met630*f* and Met803r [28], *L. reuteri* by using species specific primers [29] and total bacteria by using the primers Eub338 (`ACTCCTACGGGAGGCAGCAG`) and Eub518 (`ATTACCGCG GCTGCTGG`) [30, 31]. Before quantification, qPCRs were done with DNA dilutions ranging from 1:10, 1:20, 1:100, 1:500, and 1:1 000 including all primer sets in order to determine the dilution factor required for diminishing inhibitory factors, whilst remaining within the range of standard curve. Quantification of all DNA samples was performed in duplicates using two DNA dilutions (determined before as appropriate) using a Bio-Rad iQ5 multicolour real-time PCR detection system and the IQ$^{TM}$ SYBR$^{®}$ Green Supermix (Bio-Rad laboratories, Inc.). qPCR reactions were set up to a final volume of 20 μL containing the following components: 2x IQ$^{TM}$ SYBR$^{®}$ Green Supermix, 10 μM each forward and reverse primers, 3 μL DNA template and 5 μL milliQ water. Four to six non-template controls were included in each assay. Plasmid-coded partial 16S rRNA gene originating from methanogens (*Methanoculleus bourgensis*), bacteria (*Escherichia coli*) and *L. reuteri*, respectively were used as standard curves and applied in $10^8$ to $10^1$ copy numbers. The program used was as follows: initial temperature 95˚C for 7 minutes, followed by 40 cycles at 95˚C for 40 s, 60˚C (*L. reuteri* 64˚C) for 60 s (*L. reuteri* 30 s) and 72˚C for 40 s (*L. reuteri* 30 s). The specificity of the PCR product was estimated by melting curve analysis, which consisted of 50 gradual denaturation cycles. The temperature range was set from 55 to 95˚C, dwelled 10 s and increased 0.5˚C in each cycle. PCR products were additionally checked by gel electrophoresis. The data generated were collected and analyzed with Bio-Rad iQ5 standard edition optical system software (version 2.0), from which sorted data were exported to Microsoft Excel for further analysis. Final values can be found in the supplementary file.

## Preparation of libraries for Illumina amplicon sequencing

16S rRNA amplicon libraries were constructed as triplicates using two consecutive PCR procedures as described in [32]. The first PCR simultaneously targeted the V4 region of both bacteria and archaea, using the primers 515F (`ACACTCTTTCCCTACACGACGCTCTTCCGATCTN NNNGTGBCAGCMGCCGCGAA`) and 805R (`AGACGTGTGCTCTTCCGATCTGGACTACHVGG GTWTCTAAT`) and attaches adaptors to the amplicons [32]. The reaction mixture contained 2x Phusion High-Fidelity DNA Polymerase/dNTP mix (Thermo Fischer Scientific, Hudson, NH, USA), 10 μM of each primer, and approx. 10 ng DNA template in a final volume of 25 μL. The condition for amplification was as following: initial denaturing at 98˚C for 30 s, 20 cycles of 10 s at 98˚C, 30 s at 60˚C, 4 s at 72˚C, and a final extension at 72˚C for 2 min. The PCR products were checked for size and quality by electrophoresis. Amplicons were purified using Agencourt AMPure XP (Becker Coulter, Brea, CA, USA), using a magnetic particle/DNA volume ratio of 0.8:1. In the second PCR, Illumina-compatible barcodes were added to the amplicons. The PCR reaction contained 10 μL purified amplicon from the first step, 2x Phusion High-Fidelity DNA Polymerase/dNTP mix and 10 μM each of the primers 5′-`AATGATACGGCGAC CACCAGATCTACACX8ACACTCTTTCCCTACACGACG`-3′ and 5′-`CAAGCAGAAGACGGC ATACGAGATX8GTGACTGGAGTTCAGACGTGTGCTCTTCCGATCT`-3′, where X8 in the primer sequence represented a specific Illumina-compatible barcode (Eurofins Genomics). The following conditions were used for the second PCR step: initial denaturing at 98˚C for 30 s, 8 cycles of 10 s at 98˚C, 30 s at 62˚C, 5 s at 72˚C, and a final extension at 72˚C for 2 min. The PCR products were checked by electrophoresis and purified using Agencourt AMPure XP. The PCR products were then each diluted to a DNA concentration of approx. 30 nM and pooled together. Pair-end sequencing was performed on the MiSeq platform (Illumina, Inc., San Diego, CA, USA) at Eurofins GATC Biotech GmbH (Konstanz, Germany) resulting in an average of 110 000 raw reads per sample. Due to poor read quality the samples H, 4˚C 24h and

H, 4˚C 48h have not been considered for further analyses. Illumina adapters and primers have been trimmed away using Cutadapt version 2.2 [33]. All reads shorter than 250 base-pairs (bp), longer than 300 bp or untrimmed were discarded. Amplicon sequence variants, abundancies and taxonomic affiliation were determined using the package *dada2* (version 1.6.0) [34] in R (version 3.4.0), which is implemented on the SLUBI computing cluster in Uppsala (running on CentOS Linux release 7.1.1503; module handling by Modules based on Lua: Version 6.0.1; https://www.slubi.se/). Trimming and filtering was jointly performed on paired-end reads. Low quality reads were removed by setting the maximum number of expected errors to 2. The remaining sequence reads were denoised, dereplicated, merged and checked for chimeras in package *dada2* according to the DADA2 pipeline tutorial (https://benjjneb.github.io/dada2/tutorial_1_8.html). Taxonomic classification of the 16S ribosomal RNA sequence variants and a phylogenetic tree were obtained by using the Silva taxonomic training dataset v132 formatted for DADA2 (https://zenodo.org/record/1172783/files/silva_nr_v132_train_set.fa.gz; https://zenodo.org/ record/1172783/files/silva_species_assignment_v132.fa.gz). A phyloseq object was created consisting of the taxonomy table and the OTU table and used for the visualization with package *phyloseq* (version 1.30.0) [35]. The sequence variants were extracted and a phylogenetic tree was generated using default parameters in FastTree (version 2.1.0) [34]. Sample metadata and phylogenetic tree were merged with the phyloseq object and visually analyzed by using the package *phyloseq* and *ggplot2* (version 3.2.1) [35] in R Studio version 3.5.2 (http://www.r-project.org). Weighted Unifrac distances were calculated by function Unifrac in package *phyloseq* and used to plot the ordination in a Principal coordinate analysis plot with the function cmdscale in package *stats* (version 3.6.2) in R core packages. Experimental environmental factors were fitted to the ordination plot by using function envfit in package *vegan* (version 2.5.6) [36]. The sequencing raw data has been submitted to NCBI sequence read archive (SRA) under the BioProject accession number PRJNA609715.

## Analysis of short chain fatty acids (SCFA)

SCFA has been analyzed using nuclear magnetic resonance (NMR) spectroscopy. Between 200–250 mg stool sample was diluted with 0.75 mL sodium phosphate buffer (0.4 M, pH 7.0), homogenized by using a vortex and centrifuged at 6300 rpm at 4˚C for 15 min. From the supernatant, 0.75 mL of fecal water has been transferred to a fresh tube and subjected to centrifugation at 20 000 ×g at 4˚C for 15 min. This step was repeated once with 600 µL of supernatant recovered from the previous centrifugation. Finally, 525 µL of the supernatant was mixed with 45 µL D2O, and 30 µL internal standard. Each sample solution (560 µL) was transferred to a 5 mm NMR tube, and the 1H NMR spectra were acquired using a Bruker Avance III spectrometer operating at 600 MHz proton frequency and equipped with a cryogenically cooled probe and an auto sampler. Each spectrum was recorded (25˚C, 128 transients, acquisition time 1.8 s, relaxation delay 4 s) using a zgesgp pulse sequence (Bruker Biospin) using excitation sculpting with gradients for suppression of the water resonance. For each spectrum, 65 536 data points were collected over a spectral width of 17 942 Hz [37]. All NMR spectra were processed using Bruker TopSpin 4.1 software. The data were Fourier-transformed after multiplication by a line broadening of 0.3 Hz and referenced to internal standard peak TSP at 0.0 ppm. Baseline and phase were corrected manually. Each spectrum was integrated using Amix 3.7.3 (Bruker BioSpin GmbH, Rheinstetten, Germany) into 0.01-ppm integral regions between 0.5 and 10 ppm, in which areas between 4.52–5.06 ppm were excluded. Each integral region was referenced to the internal standard. The integral regions corresponding to short chain fatty acids were adjusted for mg of stool samples extracted and used for further statistical analysis.

Five samples were analyzed using less amount than 250 mg stool respectively. Absolute numbers and calculation can be found in the supplementary file.

## Statistics

Friedman's test for related samples was performed. A p-value of less than 0.05 was considered significant.

## Results

### Effect of storage condition on the microbial community composition

Sufficient starting material and high quality raw read data were available to compare microbial changes between immediately frozen and storage for 24 h at RT for 8 individuals; changes between frozen and storage for 48 h at RT for 10 individuals, changes between frozen and storage for 24 h at 4˚C for 7 individuals, and changes between frozen and storage for 48 h at 4˚C for 9 individuals. In order to evaluate changes in the microbial structure and composition over time both alpha and beta diversity have been analysed. The alpha diversity indices Shannon and Simpson indicated small changes in the community structure for most of the samples considering the scaling (Fig 1). Larger variations were observed for individuals E, F, H. However, the variance in alpha diversity within the triplicates was in part much higher than that observed between aliquots of different storage conditions (S1 Fig).

Principal coordinate analysis (PCoA) using the weighted UniFrac matrix was used to compare the beta diversity. Most individuals showed variations in the community composition when comparing the differently stored stool samples. Although the majority of them did not form clear clusters with regard to the different storage temperatures, the PCoA revealed little influence of time and temperature on the microbial composition, as indicated by the arrows (Fig 2).

### Effect of storage condition of absolute abundance of total bacteria, methanogens and *Lactobacillus reuteri*

Generally, only marginal variations of the 16S rRNA gene copy numbers were observed in case of all three targets, independently if frozen immediately or stored at 4˚C or RT for up to 48 h (Figs 3 and 4). The average deviations from the immediately frozen sample were between 1.6 and 2.3% (±2.5–3.2) in case of methanogens, 0.2 and 0.8% (±0.9–1.8) in case of total bacteria, and 2.0 and 3.2% (±2.2–2.8%) in case of *L. reuteri* (Table 1).

*L. reuteri* was found in sufficiently high numbers in only four out of ten samples to be above the detection limit of qPCR and to exclude inhibitory effects (Fig 4). No significant effect of storage time or temperature on the absolute abundance of methanogens or *L. reuteri* was observed (Figs 3 and 4, Table 1). A small significant difference (p = 0.024) in total bacteria (0.8%) was found between immediately frozen and RT_48h samples (S1 Appendix, Fig 3B, Table 1).

### Effect of storage condition on the SCFA acetate, propionate and butyrate

Although all participants contributing to this study received detailed instructions on the amount/volume of sample to be delivered, only five stool samples contained sufficient material to examine both the microbial community and the SCFA composition. These five stool samples were subjected to SCFA analysis using NMR. Storage at RT had a dramatic effect on levels of acetate, propionate and butyrate (S2 Fig). Already the storage for 24 hours at RT promoted

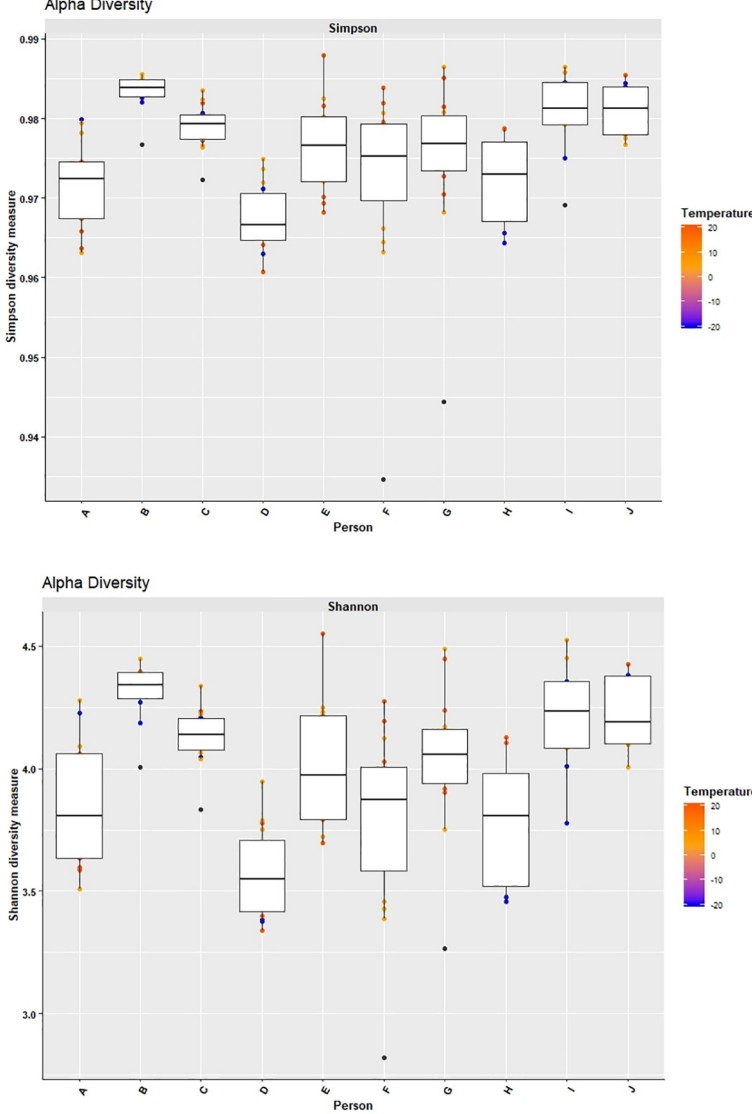

**Fig 1.** Variation in alpha diversity (Simpson (A) and Shannon (B)) displayed as box plot of individuals A-J. Extended lines indicate variability outside the upper and lower quantity of the box, whereas outliers are plotted as individual points. Blue: immediately frozen samples, orange: 4˚C, red: 20˚C.

an increase of all three SCFA to more than 100% in some cases when compared to values obtained from immediately frozen stool sample.

Acetate values partly increased by more than 150%. The mean average deviation from the initial value (immediately frozen) for acetate, propionate and butyrate were between 94 to 105% (±35%-45%) after 48 h (Table 2). However, also storage at 4˚C led to an increase of all three SCFA in four out of five samples albeit to a lesser extent (S2 Fig). In one sample, the concentration of SCFAs dropped during storage at 4˚C. The average deviation for all SCFA from initial values was between 14 and 20% (±10–19%) after 24 h at 4˚C considering both increase and decrease (Table 2). Storage up to 48 h resulted in slightly further increase of the average deviation (Table 2). However, when calculating SCFA ratios the picture changed.

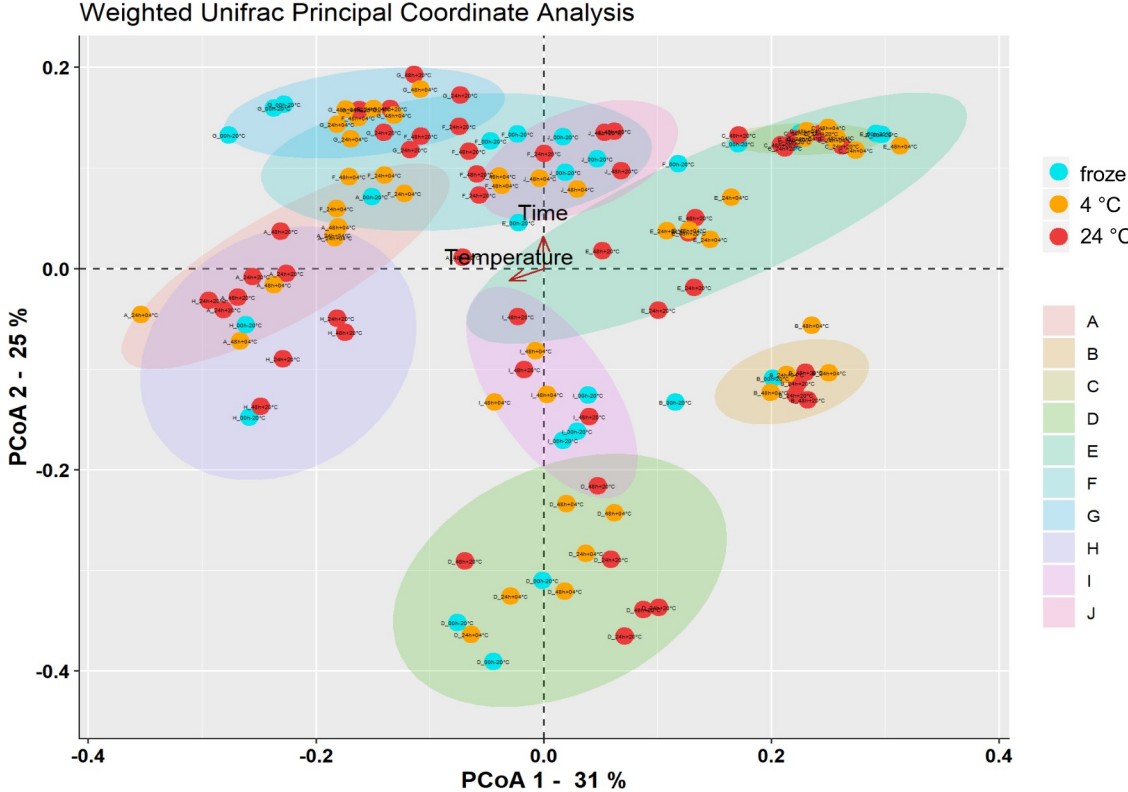

**Fig 2. PCoA plot of weighted UniFrac distance.** Principal coordinate analysis displaying beta diversity of the bacterial communities in the stool samples of 10 individuals. blue: frozen, orange: 4˚C, red: 20˚C. Ellipses indicate the individuals.

(Fig 5): For the majority of the calculated ratios the mean standard deviation remains below 20% when compared to the respective ratios obtained for the immediately frozen sample (Table 2, Fig 5, S2 and S3 Figs). The average deviation from the initial value was least pronounced in those ratios containing more than one SCFA in the divisor. The ratios acetate/(acetate+propionate+butyrate) and propionate/(acetate+propionate+butyrate) differ by less than 10% from the ratios obtained from immediately frozen samples (Table 2, Fig 5, S2 and S3 Figs). Most important and in contrast to the absolute values, the majority of the SCFA ratios indicate a much less pronounced impact of increased temperature or storage time on the SCFA levels. The observed variations in the SCFA ratios remain comparatively independent of temperature and how long a sample was stored before extraction. There was no statistically significant difference between the SCFA ratios with respect to sample handling (see Fig 5, lower panels).

## Discussion

It is generally not recommended to store fecal samples at RT or longer than 12 h at 4˚C since metabolism and proliferation of some bacteria might continue, with consequences on microbial structure and SCFA profile. Moreover, the intake of oxygen might damage strictly anaerobic microorganisms [38]. A number of attempts have been made to determine the optimal collection method to preserve microbial community and metabolic composition. Direct freezing at -80˚C is considered the reference method, however is not feasible for many settings. Instead, a -20˚C freezer is available in most homes and freezing at -20˚C may also be applied

**Fig 3. Absolute 16S rRNA gene copy numbers of total methanogens (A) and total bacteria (B) from stool samples retrieved from individuals A-J, stored at the conditions as indicated in the legend.** Frozen stands for immediately frozen at -20˚C.

[38, 39] but this requires willingness to place samples in the freezer and resources for shipping the samples in a frozen state. Several studies reported positive effects of preservation media and stabilizers [11–16]. However, even if patients are asked to describe how samples were handled before submission to the laboratory, it is difficult to have complete control over what has happened and samples, for diverse reasons, may have been exposed to room temperature for hours before reaching the laboratory. There are only few studies that looked at the effects of

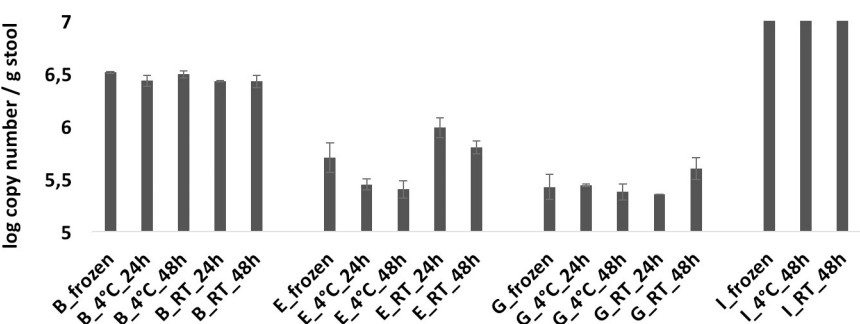

**Fig 4. Absolute 16S rRNA gene copy numbers of *L. reuteri* from stool samples retrieved from individuals B, E, G, and I, stored at the conditions as indicated in the legend.** Frozen stands for immediately frozen at -20˚C.

**Table 1. Average deviation in percent (%) of absolute 16S rRNA gene copies from initial values obtained from untreated (immediately frozen) samples.** Detailed list of values can be found in S1 Appendix.

| | 4ºC_24h | 4ºC_48h | RT_24h | RT_48h |
|---|---|---|---|---|
| Methanogens | 2.0±3.2 | 2.3±3.2 | 1.6±2.5 | 2.3±2.5 |
| Total bacteria | 0.7±1.8 | 0.2±1.6 | 0.5±0.9 | 0.8±1.5 |
| *L. reuteri* | 2.0±2.2 | 2.9±2.8 | 2.5±2.2 | 3.2±2.3 |

short-term storage conditions on unpreserved microbial community and those came to different conclusions [10, 12, 39–41]. Ott *et al* reported a significant reduction of both bacterial diversity and total number of bacteria after already 8 h at both 4˚C and RT and Cardona *et al* observed alterations in the relative abundance across most taxa when fecal sample were subjected to RT for 24 h [39, 41]. On the other hand, two studies reported only minor alterations in taxa abundances when stored at both 4˚C or RT for up to 24 h [10, 12]. Lauber *et al*. reported stability of the microbiota even for up to 14 days at 4˚C and 20˚C [40].

In the present study, we simulated the sample acquisition of a large-scale study, in which some patients reported storing stool samples for up to two days in the fridge or at room temperature before handed over to the hospital. To our knowledge, this is the first study evaluating the impact of storage time and temperature on the absolute abundance of methanogens, and of *L. reuteri* in particular. Methanogens are extremely oxygen sensitive and even trace amount of oxygen causes stress to most of methanogenic species by damaging the cell membrane and proteins [42]. However, we could demonstrate that the absolute abundance of Methanogens appeared stable at both RT and 4˚C for up to 48 h. Only small variation occurred, which did not correlate to time or temperature. The same findings were observed for *Lactobacillus reuteri*, which is facultative anaerobe and might be therefore able to proliferate under certain conditions. Alpha diversity indices and PCoA plots indicated small changes in the overall microbial community, when stored at 4˚C or RT for up to 24 h, which is in agreement with the findings of Ott and Caroll [10, 41]. We also found that even a prolonged storage up to 48 h did not significantly alter the microbial composition any further. The variations observed within the triplicates indicates that other factors including differences in sample handling, PCR and sequencing may also account for the observed alterations in the microbial community. This is further supported by the finding that the total number of bacteria remained stable over time, even if a significant decrease was found after 48 h at RT. However, the effect of this decrease was considered very small and less than 1%. This is contradictory to Ott et al who reported a dramatic reduction of total bacteria, however the different techniques used might not allow a

**Table 2. Average deviation +/- standard deviation of SCFA in percent from initial values obtained from immediately frozen samples.** Detailed list can be found in the supplementary file.

| | frozen → 4˚C_24h | frozen → 4˚C_48h | frozen → RT_24h | frozen → RT_48h |
|---|---|---|---|---|
| Acetate | 20 ±10 | 28 ±18 | 99 ±67 | 105 ±36 |
| Butyrate | 14 ±19 | 18 ±9 | 67 ±45 | 94 ±45 |
| Propionate | 15 ±13 | 21 ±12 | 76 ±30 | 102 ±38 |
| Butyrate/Acetate | 19 ±8 | 17 ±6 | 15 ±13 | 19 ±7 |
| Butyrate/Propionate | 9 ±3 | 12 ±10 | 30 ±28 | 18 ±20 |
| Butyrate/(Acetate+Propionate) | 18 ±7 | 15 ±6 | 14 ±12 | 16 ±6 |
| Acetate/(Acetate+Propionate+Butyrate) | 3 ±1 | 3 ±2 | 3 ±3 | 4 ±1 |
| Propionate/(Acetate+Propionate+Butyrate) | 7 ±2 | 7 ±4 | 10 ±8 | 9 ±4 |
| Butyrate/(Acetate+Propionate+Butyrate) | 17 ±7 | 15 ±6 | 14 ±11 | 15 ±6 |

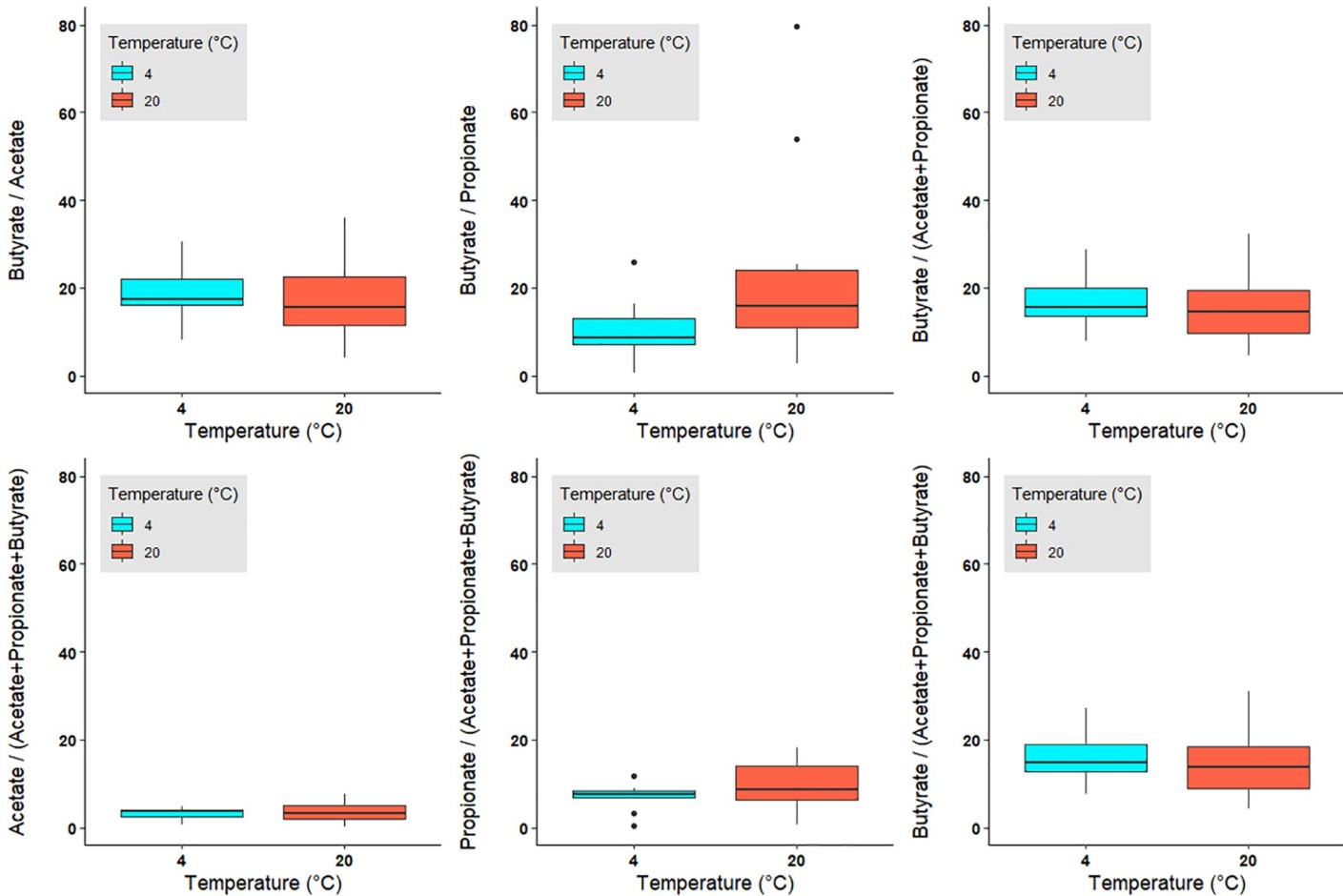

**Fig 5. Deviation of SCFA ratios in % from those ratios obtained from the immediately frozen samples displayed as box plots.** Blue: samples stored at 4˚C up to 48 h, red: samples stored at 20˚C up to 48 h. Extended lines indicate variability outside the upper and lower quantity of the box, whereas outliers are plotted as individual points. Ratios are also displayed in S2 and S3 Figs. Calculations and values can be found in the supplementary file (S1 Appendix).

direct comparison of the results [41]. Although the composition and structure of the microbial community has only changed little over time, the community still appeared to be active, especially at RT. The absolute levels of acetate, propionate and butyrate increased dramatically within 24 hours, indicating general metabolic activities. Even storage at 4˚C could not completely suppress metabolic activities, but proved to be clearly beneficial. Interestingly enough, the effect of time and temperature was strongly diminished when looking at ratios instead of absolute values. Obviously, the SCFAs increase almost proportionally with time at both 4˚C and in RT, so that the factors time and temperature became irrelevant. This means that samples collected cool or kept at RT can be compared to each other, but not with immediately frozen samples. The deviations from the immediately frozen samples were only negligible for the ratio acetate/(acetate+propionate+butyrate). A direct comparison with frozen samples should only be considered taking into account the threshold values identified here as mean deviations. Several other studies show that the immediate preservation by e.g. ethanol or RNA-later can replace an immediate freezing of the samples [15, 43]. However, this may not always be feasible. The results found here can thus support those studies that cannot implement immediate freezing or stabilizing methods, but still want to integrate metabolomics.

## Conclusions

By analyzing the impact of storage time and temperature, we conclude that realistic values for the absolute abundance of methanogens, *L. reuteri* and total bacteria can still be obtained even after storage for up to 48 hours at RT or 4°C. The overall microbial composition appears to be influenced more by laboratory error introduced during sample processing than by the actual effects of storage temperature and time. Although significant microbial activities have been demonstrated at both 4°C and RT, the SCFAs may be considered as long as these are taken into account as ratios and originated from similar storage conditions.

## Supporting information

**S1 Fig. Alpha diversity (Simpson and Shannon) of the replicates displayed as box plot at the different time points and temperatures analyzed.** Individuals are indicated by different colors and capitals A-J. Extended lines indicate variability outside the upper and lower limits of the box, whereas outliers are plotted as individual points.
(JPEG)

**S2 Fig. Effect of time and temperature on SCFA concentrations and SCFA ratios displayed as average deviation from the initial values obtained from immediately frozen samples.**
(PDF)

**S3 Fig. Deviation of SCFA ratios in % from those ratios obtained from the immediately frozen samples after 24 h and 48h displayed as box plots.** Blue: samples stored at 4°C, red: samples stored at 20°C. Extended lines indicate variability outside the upper and lower quantity of the box, whereas outliers are plotted as individual points.
(TIFF)

**S1 Appendix. Data obtained from short chain fatty acid and qPCR analyses can be found in the supplementary file "S1_Appendix".**
(XLSX)

## Author Contributions

**Conceptualization:** Janet L. Cunningham, Annica J. Rasmusson, Bettina Müller.

**Data curation:** Ludvig Bramstång, Shishanthi Jayarathna.

**Formal analysis:** Ludvig Bramstång, Shishanthi Jayarathna, Bettina Müller.

**Funding acquisition:** Janet L. Cunningham.

**Investigation:** Ludvig Bramstång.

**Project administration:** Janet L. Cunningham.

**Resources:** Annica J. Rasmusson, Ali Moazzami.

**Supervision:** Janet L. Cunningham, Bettina Müller.

**Visualization:** Abhijeet Singh, Bettina Müller.

**Writing – original draft:** Ludvig Bramstång, Bettina Müller.

**Writing – review & editing:** Janet L. Cunningham, Abhijeet Singh, Shishanthi Jayarathna, Annica J. Rasmusson, Ali Moazzami, Bettina Müller.

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
