## [Decision Letter · Decision Letter 0]

30 Apr 2020

PONE-D-20-07045

Impact of time  and temperature  on gut microbiota and SCFA composition in stool samples

PLOS ONE

Dear Dr. Müller,

Thank you for submitting your manuscript to PLOS ONE. After careful consideration, we feel that it has merit but does not fully meet PLOS ONE’s publication criteria as it currently stands. Therefore, we invite you to submit a revised version of the manuscript that addresses the points raised during the review process.

We would appreciate receiving your revised manuscript by Jun 14 2020 11:59PM. To enhance the reproducibility of your results, we recommend that if applicable you deposit your laboratory protocols in protocols.io, where a protocol can be assigned its own identifier (DOI) such that it can be cited independently in the future. For instructions see: http://journals.plos.org/plosone/s/submission-guidelines#loc-laboratory-protocols

We look forward to receiving your revised manuscript.

Kind regards,

Juan J Loor

Academic Editor

PLOS ONE

Journal Requirements:

2. Thank you for including the following consent information in the ethics statement of your manuscript:

'Control subjects gave verbal informed consent for their donated samples to be analyzed anonymously with the purpose of method development and neither samples nor data cannot be traced back to control individuals. The Regional Ethics Committee waived the need for consent in this case in accordance with Swedish law'

Please specify whether the consent was given verbally by control subjects or waived by your local ethics committee.

Additional Editor Comments (if provided):

Reviewers' comments:

Reviewer's Responses to Questions

**Comments to the Author**

1. Is the manuscript technically sound, and do the data support the conclusions?

Reviewer #1: Partly

2. Has the statistical analysis been performed appropriately and rigorously? 

Reviewer #1: No

3. Have the authors made all data underlying the findings in their manuscript fully available?

Reviewer #1: Yes

4. Is the manuscript presented in an intelligible fashion and written in standard English?

Reviewer #1: Yes

5. Review Comments to the Author

Reviewer #1: In this article, the authors want to verify the effect of delaying the freezing of the samples for conservation on the composition of the microbiome and its related metabolism. To do so, they use qPCR to measure the abundance of some low abundance species, which could have an effect on health (methanogens and Lactobacillus reuteri). They also use 16S sequencing to measure the alpha and beta diversity of samples and NMR spectroscopy to measure 3 different short-chain fatty acids (SCFA), which are known to have a role in the gut-brain axis. The sample size is low but corresponds to the size used by similar studies on the effect of storage and stabilizers. The use of three different analysis is a good procedure to show the effect of storage protocols on the microbiome at various levels. The lack of any statistical analysis is a bit concerning in view of the large standard deviations seen in most comparisons. Overall, it is an analysis of interest for the community at large but still needs some work to be more convincing.

Major comments :

1. The discussion and results talk about small variations and changes in the levels of various measures but no statistical analysis is presented to support those variations. Why no statistical tests were realized at any point in the analysis to verify if the observed differences were significant or not?

2. I understand that samples were obtained from two different groups of subjects with different types of consent forms and that those “patients” are relevant to your ongoing large-scale study. However, why use two groups of participant (patient and control) in this study when these are never discussed in the paper and that the selected patients have affective disorders whose effect on the microbiome is not discussed?

3. In the sample management section (lines 131 to 139), it is mentioned that there are 5 different storage conditions used on all the samples but for some of the samples or some of the analysis, some of the sample/condition pairs are missing. The absence of 24h treatment data for both temperature for sample I and J is explained in the methods (line 150) but sample H had no results for 4°C in figure 2 and supplementary figure 1 but has results in the rest of the analysis?

4. Lactobacillus reuteri is a bacterium with many potential positive effects on health due to its antimicrobial activities and its effect on the reduction of pro-inflammatory cytokines production. However, since it is detected in less than half of the samples, other low abundance bacteria other than L. reuteri should have been used in the measures to support the claims.

5. Figure 3 and 4 show the gene copy numbers of each of the measured bacteria but these graphs are hard to see and analyze. In my opinion, the y-axis should not start at 0, but should rather zoom into the top part of the graph so that we can more precisely see the variations between the bars.

6. At lines 249 and 313, it is mentioned that only 5 samples were used in the SCFA analysis instead of all 10. Why were only 5 samples used and how were they selected?

7. In most of the analysis, the 24h and 48h samples are separated. Why is there no distinction made between the 24h and 48h samples in figures 1 and 5?

Minor comments :

1. Line 75, missing word between “subjects” and “may”, I suggest “which”.

2. Line 83, either “Although” or “but” should be removed.

3. Line 118, the “and” should be replaced by a “,”.

4. Line 128, “cannot” should be “can”.

5. Line 149-150, the condition in which only a duplicate was purified should be noted.

6. Line 345, “Calculation and numbers” could be replaced with “Calculations and values”.

7. Line 348, replace “or” with “for”.

8. In S1_Appendix “difference” is written as “differenz” in the header of sheet “SCFA_Difference to frozen in %”.

9. When opening S1_Appendix, there is a request to obtain updated values for files on the authors computer.

6. PLOS authors have the option to publish the peer review history of their article (what does this mean?). If published, this will include your full peer review and any attached files.

Reviewer #1: No

---

## [Author Response · Author response to Decision Letter 0]

12 Jun 2020

Dear Editor.

Please find below the response to the reviewer`s comments. We also improved the quality of some figures by loading them up separately as e.g. Figure_1_A and Figure_1_B (before Figure_1).

Additional changes not motivated by the reviewer:

Line 318: Table 1: “Mean average deviation” changed to “Average deviation”

Line 155: abbreviation of room temperature =RT has been introduced 

Response to the reviewer comments

Major comments:

1. The discussion and results talk about small variations and changes in the levels of various measures but no statistical analysis is presented to support those variations. Why no statistical tests were realized at any point in the analysis to verify if the observed differences were significant or not?

In case of overall changes of the microbial community, statistical measures have been applied to produce the PCoA plot in figure 2 as indicated by the arrows representing impact of time and temperature. Furthermore, we added analysis using Friedman’s test for related samples to test if there was a significant difference between the qPCR data of methanogens, total bacteria and Lactobacillus reuteri as well as the SCFA ratios. The following changes in the method part under a new header “Statistics” and in the results part have been added: 

Line 274-276: “Friedman’s test for related samples was performed. A p-value of less than 0.05 was considered significant.”

Line 287: “..considering the scaling..”

Line: 334-336: “A small significant difference (p=0.024) in total bacteria (0.8%) was found between immediately frozen and RT_48h samples (S1_Appendix, Figure 3B, Table 1).”

Line 367: “significantly lower” has been changed to “..to a lesser extent.”

Line 433-434: “..,even if a significant decrease was found after 48 h at RT. However, the effect of this decrease was considered very small and less than 1 %.” 

2. I understand that samples were obtained from two different groups of subjects with different types of consent forms and that those “patients” are relevant to your ongoing large-scale study. However, why use two groups of participant (patient and control) in this study when these are never discussed in the paper and that the selected patients have affective disorders whose effect on the microbiome is not discussed?

We thank the reviewer for this note, and agree that for the actual question of the project it was irrelevant whether it was a patient or a volunteer. Since the distinction between control and patient is actually misleading and also irrelevant for the key message of the study presented. We have rewritten the method section (line 118-130) accordingly. The word "controls" has been removed and only the recruitment process has been described.

3. In the sample management section (lines 131 to 139), it is mentioned that there are 5 different storage conditions used on all the samples but for some of the samples or some of the analysis, some of the sample/condition pairs are missing. The absence of 24h treatment data for both temperature for sample I and J is explained in the methods (line 150) but sample H had no results for 4°C in figure 2 and supplementary figure 1 but has results in the rest of the analysis?

In three cases the material was not enough to perform DNA triplicates or to expose the stool samples to all storage conditions, as mentioned in the method section and in the results part. 

We have modified the sentence and indicated which samples are affected: 

Line 168: “Out of these three, one was purified in duplicates for all conditions (sample B), and two were purified in triplicates but covering only storage condition 1, 4 and 5 (sample I, and J). “ 

Although Illumina sequencing was performed in triplicates, there were a few samples which we sorted out due to poor read quality and which we mentioned in the results part. This includes sample H at 4°C, 24h and 48h. However, we see that this was not stated clearly. We added the following sentence in the method section: Line 222: “Due to poor read quality the samples H, 4°C 24h and H, 4°C 48h have not been considered for further analysis.”

4. L. reuteri is a bacterium with many potential positive effects on health due to its antimicrobial activities and its effect on the reduction of pro-inflammatory cytokines production. However, since it is detected in less than half of the samples, other low abundance bacteria other than L. reuteri should have been used in the measures to support the claims.

We agree with the reviewer on this point. In this case, however, we were particularly interested in the effect of temperature and time on the absolute abundance of L. reuteri, since this bacterium has been described as important for depression and mood disorders and is also considered in many other studies due to the positive effects on health as mentioned by the reviewer. We expected that not all samples would have sufficiently high L. reuteri levels to be detected using qPCR. This was also one reason why we decided to recruit a more heterogeneous group of subjects including both patients and healthy volunteers. Even if the number of samples is statistically very small, the result can still be regarded as significant for L. reuteri, since we observed only very slight deviations at 24 ° C or 4 ° C over time in all four stool samples. 

5. Figure 3 and 4 show the gene copy numbers of each of the measured bacteria but these graphs are hard to see and analyze. In my opinion, the y-axis should not start at 0, but should rather zoom into the top part of the graph so that we can more precisely see the variations between the bars.

We see the point. The Figures 3 and 4 have been adjusted as recommended by the reviewer.

6. At lines 249 and 313, it is mentioned that only 5 samples were used in the SCFA analysis instead of all 10. Why were only 5 samples used and how were they selected?

This was due to a lack of stool samples, and represents a situation that should not be neglected when planning larger studies, as it can easily reduce the number of controls and patients. We added to the section “Results, Effect of storage condition on the SCFA acetate, propionate and butyrate” the following sentence:

Line 344-346: “Although all participants contributing to this study received detailed instructions on the amount/volume of sample to be delivered, only five stool samples contained sufficient material to examine both the microbial community and the SCFA composition.” 

In three cases the material was not enough to perform DNA triplicates or to expose the stool samples to all storage conditions “See also comment 3”. 

6. In most of the analysis, the 24h and 48h samples are separated. Why is there no distinction made between the 24h and 48h samples in figures 1 and 5?

Figure 1: We found it more convenient for the reader to capture changes over time at one glimpse, and only separated the temperature by colors. However, the reviewer is right, that distinguishing the different time points might be of interest for the reader. In Figure S1 we distinguish between 24h and 48.

Figure 5: As we separated between 24h and 48h in Table 2, we found it more visible to catch the overall changes in SCFA ratios up to 48 h at either 4°C or 20°C at one glimpse. However, we have added an additional figure in the supplementary for the convenience of the reader:

Figure S3: Deviation of SCFA ratios in % from those ratios obtained from the immediately frozen samples after 24 h and 48h displayed as box plots. Blue: samples stored at 4 °C, red: samples stored at 20 °C. Extended lines indicate variability outside the upper and lower quantity of the box, whereas outliers are plotted as individual point.

Minor comments:

1. Line 77, missing word between “subjects” and “may”, I suggest “which”.

Now line 76: Added “which”

2. Line 83, either “Although” or “but” should be removed.

Now line 84: Removed “but”

3. Line 118, the “and” should be replaced by a “,”.

Now line 120-121: “and” has been replaced by “,”

4. Line 128, “cannot” should be “can”.

Changed to “can”, now in line 127

5. Line 149-150, the condition in which only a duplicate was purified should be noted.

Information has been added (now line 169-170), see also comment 3, 

6. Line 345, “Calculation and numbers” could be replaced with “Calculations and values”.

“Numbers” has been replaced by “values”. 

7. Line 348, replace “or” with “for”.

“or “ is right here

8. In S1_Appendix “difference” is written as “differenz” in the header of sheet “SCFA_Difference to frozen in %”.

The spelling error has been corrected, wherever it appeared in the appendix file.

9. When opening S1_Appendix, there is a request to obtain updated values for files on the authors computer.

This has been changed: The file can be opened with “only read” options. No updated values will be requested when opening.

---

## [Decision Letter · Decision Letter 1]

17 Jul 2020

Impact of time  and temperature  on gut microbiota and SCFA composition in stool samples

PONE-D-20-07045R1

Dear Dr. Müller,

We’re pleased to inform you that your manuscript has been judged scientifically suitable for publication and will be formally accepted for publication once it meets all outstanding technical requirements.

Kind regards,

Juan J Loor

Academic Editor

PLOS ONE

Additional Editor Comments (optional):

Reviewers' comments:

Reviewer's Responses to Questions

**Comments to the Author**

1. If the authors have adequately addressed your comments raised in a previous round of review and you feel that this manuscript is now acceptable for publication, you may indicate that here to bypass the “Comments to the Author” section, enter your conflict of interest statement in the “Confidential to Editor” section, and submit your "Accept" recommendation.

Reviewer #1: All comments have been addressed

2. Is the manuscript technically sound, and do the data support the conclusions?

Reviewer #1: Yes

3. Has the statistical analysis been performed appropriately and rigorously? 

Reviewer #1: Yes

4. Have the authors made all data underlying the findings in their manuscript fully available?

Reviewer #1: Yes

5. Is the manuscript presented in an intelligible fashion and written in standard English?

Reviewer #1: Yes

6. Review Comments to the Author

Reviewer #1: All our comments were addressed properly by the authors. We are satisfied with the new version of the manuscript Thanks.

7. PLOS authors have the option to publish the peer review history of their article (what does this mean?). If published, this will include your full peer review and any attached files.

Reviewer #1: No

---

## [Editor Report · Acceptance letter]

23 Jul 2020

PONE-D-20-07045R1 

Impact of time and temperature on gut microbiota and SCFA composition in stool samples  

Dear Dr. Müller:

I'm pleased to inform you that your manuscript has been deemed suitable for publication in PLOS ONE. Congratulations! Your manuscript is now with our production department. 

Kind regards, 

on behalf of

Dr. Juan J Loor 

Academic Editor

PLOS ONE